# SARS due to COVID-19: Predictors of death and profile of adult patients in the state of Rio de Janeiro, 2020

Tatiana de Araujo Eleuterio[1,2]*, Marcella Cini Oliveira[3], Mariana dos Santos Velasco[2], Rachel de Almeida Menezes[2], Regina Bontorim Gomes[2], Marlos Melo Martins[4], Carlos Eduardo Raymundo[1], Roberto de Andrade Medronho[1,3]

1 Institute of Studies in Public Health, Federal University of Rio de Janeiro, Rio de Janeiro, State of Rio de Janeiro, Brazil, 2 Faculty of Nursing, State University of Rio de Janeiro, Rio de Janeiro, State of Rio de Janeiro, Brazil, 3 Faculty of Medicine, Federal University of Rio de Janeiro, Rio de Janeiro, State of Rio de Janeiro, Brazil, 4 Department of Pediatrics, Institute of Child Care and Pediatrics Martagão Gesteira, Federal University of Rio de Janeiro, Rio de Janeiro, State of Rio de Janeiro, Brazil

* tatirodriguesaraujo@yahoo.com.br

## Abstract

### Introduction

We aimed to describe the profile of adult patients and analyze the predictors of death from severe acute respiratory syndrome (SARS) due to coronavirus disease 2019 (COVID-19) in the state of Rio de Janeiro. Knowledge of the predictors of death by COVID-19 in Rio de Janeiro, a state with one of the highest mortality rates in Brazil, is essential to improve health care for these patients.

### Methods

Data from the Information System for Epidemiological Surveillance of Influenza and the Mortality Information System were used. A binary logistic regression model evaluated the outcome of death, sociodemographic data, and clinical-epidemiological and health care covariates. Univariate, bivariate, and multivariate statistics were performed with the R program, version 4.0.0.

### Results

Overall, 51,383 cases of SARS due to COVID-19 among adults were reported in the state between March 5 and December 2, 2020. Mortality was high (40.5%). The adjusted final model presented the following predictors of death in SARS patients due to COVID-19: male sex (odds ratio [OR] = 1.10, 95% confidence interval [CI], 1.04–1.17); age (OR = 5.35, 95% CI, 4.88–5.88; $\geq$75 years); oxygen saturation <95% (OR = 1.48, 95%CI, 1.37–1.59), respiratory distress (OR = 1.31, 95%CI, 1.21–1.41) and dyspnoea (OR = 1.25, 95%CI, 1.15–1.36), the presence of at least one risk factor/comorbidity (OR = 1.32, 95%CI, 1.23–1.42), chronic kidney disease (OR = 1.94, 95%CI, 1.69–2.23), immunosuppression (OR = 1.51, 95%CI, 1.26–1.81) or chronic neurological disease (OR = 1.36, 95%CI, 1.18–1.58), and ventilatory support, invasive (OR = 8.89, 95%CI, 8.08–9.79) or non-invasive (OR = 1.25, 95%CI, 1.15–1.35).

**Data Availability Statement:** All relevant data are within the paper and its Supporting Information files.

**Funding:** The Carlos Chagas Filho Foundation for Research Support of the State of Rio de Janeiro (http://www.faperj.br/) FAPERJ was responsible for granting a researcher scholarship to co-author RAM, (Process E_18/2015TXB): AÇÃO EMERGENCIAL COVID-19 - Chamada A - Apoio a Rede de Pesquisa em Vírus Emergentes e Reemergentes. Rio de Janeiro, RJ, Brazil. The National Council for Scientific and Technological Development (CNPq) (https://www.gov.br/cnpq/pt-br) and the Federal University of Rio de Janeiro (UFRJ) are responsible for granting the Scientific Initiation Scholarship to co-author MCO, through the Institutional Program for Scientific Initiation Scholarships (PIBIC). The National Council for Scientific and Technological Development (CNPq) (https://www.gov.br/cnpq/pt-br) and the Department of Training and Support for the Formation of Human Resources (DCARH) of the State University of Rio de Janeiro (UERJ) are responsible for granting the Scientific Initiation Scholarship to co-author MSV, through the Institutional Program of Scientific Initiation Scholarships (PIBIC).

**Competing interests:** The authors have declared that no competing interests exist.

**Abbreviations:** COVID, 19—Coronavirus Disease 2019; ICU, intensive care unit; MERS, Middle East Respiratory Syndrome; OR, odds ratio; ROC, receiver operating characteristic; SARS, Severe Acute Respiratory Syndrome; SARS-CoV-2, Severe Acute Respiratory Syndrome Coronavirus 2; SIM, Mortality Information System; SIVEP- Gripe, Information System for Epidemiological Surveillance of Influenza; VIF, variance inflation factor; WHO, World Health Organization (WHO).

## Conclusions

Factors associated with death were male sex, old age, oxygen saturation <95%, respiratory distress, dyspnoea, chronic kidney and neurological diseases, immunosuppression, and use of invasive or noninvasive ventilatory support. Identifying factors associated with disease progression can help the clinical management of patients with COVID-19 and improve outcomes.

## Introduction

Similar to the severe acute respiratory syndrome (SARS) (2002/2003) and Middle East respiratory syndrome coronavirus (MERS) (2012) epidemics, the current coronavirus disease 2019 (COVID-19) pandemic presents critical challenges to public health and the scientific community [1]. The SARS-CoV-2 virus causes COVID-19, which results in flu-like symptoms or evolves into severe forms that characterize SARS. On January 30, 2020, the World Health Organization (WHO) declared the disease a public health emergency of international interest; on March 11, 2020, the WHO declared COVID-19 a pandemic.

Since March 20, 2020, the Brazilian Ministry of Health has verified the community transmission of SARS-CoV-2 throughout the national territory; further, mitigation measures were adopted to control the epidemic. As of 31 October 2021, the WHO recorded 627,104,342 confirmed cases of COVID-19, with 6,567, 552 deaths [2]. In Brazil, 34,815,258 new cases and 687,962 deaths from the disease were confirmed in the same time frame. A total of 2,103,714 patients with SARS caused by COVID-19 were hospitalized in Brazil between February 26, 2020 and October 22, 2022 [3].

In 2000, the Ministry of Health implemented the Information System for Epidemiological Surveillance of Influenza (SIVEP-Gripe). Initially, it emerged as a nationwide influenza surveillance system, including surveillance of influenza syndrome (SG) in Sentinel Units. The objective of SIVEP-Gripe was to identify respiratory viruses circulating in the country and monitor the demand for care for SG. Since the H1N1 influenza pandemic in 2009, influenza surveillance has reported SARS- and influenza-related deaths globally. The SIVEP-Gripe also includes surveillance of patients with SARS and SG in intensive care units [4].

This study aimed to describe the clinical and epidemiological profile of patients with SARS in the state of Rio de Janeiro, determine the predictors of death due to COVID-19-related SARS, and identify whether mortality patterns vary according to sociodemographic, clinical-epidemiological, and health care variables. This study analyzed a comprehensive period, considering all severe cases of COVID-19 in adults and performing a linkage between the Mortality Information System and the SIVEP-Gripe, seeking greater completeness of information about the outcome of death. In addition, it focused on Rio de Janeiro, a state with one of the highest mortality rates in Brazil, a country that has stood out as one of the epicenters of the pandemic [3].

## Methods

### Design

This was a cross-sectional analytical study, which evaluated the association between the sociodemographic, clinical-epidemiological, individual covariates, health care, and outcomes (death) of adult patients with SARS in the state of Rio de Janeiro, using data from the SIVEP-Gripe and the Mortality Information System (SIM).

## Data collection

Individual information on cases of SARS in the state of Rio de Janeiro was obtained from SIVEP-Gripe. Probabilistic linkage was performed with the SIM database for completeness of the outcome information (death). The key fields 'name', 'mother's name', and 'date of birth' were used to pair the probabilistic relationship. The study includes all reported cases of SARS among adults in the state of Rio de Janeiro from March 5 (the first confirmed case in the state of RJ) to December 2, 2020.

Immediate notification of SARS cases was performed by sending the scanned investigation forms uploaded to the SIVEP-Gripe. The individual covariates included in the study were sociodemographic (age, sex, race/skin color), clinical-epidemiological (signs and symptoms and comorbidities), and health care characteristics (hospitalization, use of antiviral therapies, use of intensive care, use of ventilatory support, imaging examination—X-ray and chest tomography, final classification and closing criteria). The dependent variable was death due to COVID-19 SARS or cure (non-death).

## Data analysis

Microsoft Excel 2016 (Microsoft Corporation, Redmond, WA, USA) was used for database management. Univariate, bivariate, and multivariate statistics were performed with the R program, version 4.0.0 (R Foundation for Statistical Computing, Vienna, Austria). For bivariate description, the behavior of the covariates concerning death was observed through contingency tables for categorical covariates. Pearson's chi-square test was used to evaluate the dependence of the covariates for the outcome death (yes or no). Wilcoxon test was used to evaluate the difference in age according to the outcomes. The level of significance for $\alpha$ was set at 5% for all statistical tests.

Along with descriptive analyses, the missing data in the database were evaluated according to the choice of covariates to be included in the model. Thus, depending on the variable, its clinical and epidemiological relevance, and the percentage of missing data, one of the following decisions was made: exclusion of the variable, exclusion of individuals with missing values, or creation of a specific category for the missing values.

Then, multivariate analysis was performed by constructing a binary logistic regression model. This analysis was performed only for cases of COVID-19 SARS. Using the final model, the odds ratio (OR) and respective 95% confidence intervals (95% CI) were calculated to quantify each covariate's effect on the outcome–deaths due to COVID-19 SARS. The model performance measure included the C-statistic, which is the area under the receiver operating characteristic (ROC) curve. This C-statistic was computed for the individual domain model and the final model.

Data from the Information System for Epidemiological Surveillance of Influenza and the Mortality Information System are available in electronic databases. The database was made available by the authors in Supporting Information section.

The research proposal was submitted to the Research Ethics Committee of the University Hospital Clementino Fraga Filho, Federal University of Rio de Janeiro (3981744/2020). It was exempted from ethical consideration as the research relied on secondary databases with aggregated information and without disclosing individual identification of the subjects.

## Results

Between March 5, 2020, and December 2, 2020 (epidemiological weeks 10 and 49), 51,383 cases of COVID-19 SARS among adults were reported in the State of Rio de Janeiro, and 20,785 deaths were recorded, using the linkage of the SIVEP-Gripe and SIM databases. The

peak of notification occurred in epidemiological week 19, and peak of symptom onset in week 18.

Table 1 presents the general profile of all COVID-19 SARS adult patients. There was a high frequency of patients in the 50–64 years old age group (27.7%), male patients (55.4%), and non-white patients (33.9%), and for a large proportion of patients, race/skin color was not known (36.9%). The mean age of the included patients was 62.76 years (SD ± 16.98), with a median of 64 years. The important signs and symptoms were cough, fever, dyspnoea, respiratory distress, and oxygen saturation below 95%. Regarding the presence of risk factors or comorbidities, a high frequency of cardiovascular disease and diabetes mellitus was observed; 39.3% of the patients required noninvasive ventilatory support, and 16.0% required invasive support. Imaging findings were typical for COVID-19 in 28.8% of the patients. Laboratory confirmation was provided for 77.6% of the patients. The mortality was 40.5%.

Table 2 shows the results of simple logistic regression models for each covariate. In crude analysis, the following stand out as factors associated with greater odds of death: the use of invasive ventilatory support (OR: 12.938; 95% CI: 12.061–13.879), age equal to or greater than 75 years old (OR: 6.394; 95% CI: 6.039–6.77), oxygen saturation below 95% (OR: 2.368; 95% CI: 2.25–2.492) and chronic kidney disease (OR: 2.185; CI: 2.001–2.385), among others. Factors associated with lower odds of death included pregnancy (OR: 0.278; 95% CI: 0.201–0.385) and signs and symptoms such as odynophagia (OR: 0.719; 95% CI: 0.676–0.764), among others. In comparing the fit between the bivariate models, the best fits were with the ventilatory support and age group covariates.

There was a significant difference in the age distribution between death and non-death from COVID-19 (Fig 1). Deaths mainly occurred in older individuals. The Wilcoxon test result indicated a statistically significant difference between the ages of the individuals according to the outcome.

Table 3 presents the odds ratios of death and respective confidence intervals of the covariates included in the final model. Males had a higher likelihood of death than females. Regarding the age group, the higher the age group, the greater the likelihood of death. The patients who presented with oxygen saturation below 95%, respiratory distress, or dyspnoea were more likely to progress to death. The risk factors/comorbidities that were most related to death were chronic kidney disease, immunosuppression, and chronic neurological disease. Patients who received invasive or noninvasive ventilatory support had a higher likelihood of death than those who did not require ventilatory support. The C-Statistic of goodness-of-fit of the final model was 0.800, which was a significantly high value, indicating good performance.

Fig 2 shows the results of Table 3. Again, the highest odds ratio values were for the variables of age 75 years or older and use of invasive ventilatory support.

## Discussion

According to the present study, adult patients with SARS due to COVID-19 in the state of Rio de Janeiro were mostly male and aged between 50 and 64 years. The signs and symptoms of the disease included cough, fever, dyspnoea, respiratory distress, and oxygen saturation lower than 95%. There was a high frequency of individuals with at least one risk/comorbidity factor, mainly cardiovascular disease and diabetes mellitus. Mortality was high (40.5%).

The final model identified the following predictors of death due to COVID-19 SARS: male sex; old age; the presence of respiratory distress, dyspnoea, and oxygen saturation less than 95%; having at least one risk factor/comorbidity, namely chronic kidney disease, immunosuppression or chronic neurological disease; and the use of ventilatory support, whether invasive or noninvasive.

**Table 1. Profile of SARS due to COVID-19 cases, State of Rio de Janeiro, March–December 2020.**

| Variable | | COVID-19 SARS (n = 51,383) | |
|---|---|---|---|
| | | **n** | **%** |
| **Age group** | | | |
| | 18–49 years old | 11,991 | 23.3 |
| | 50–64 years old | 14,256 | 27.7 |
| | 65–74 years old | 11,383 | 22.2 |
| | 75 years or older | 13,753 | 26.8 |
| **Sex** | | | |
| | Female | 22,904 | 44.6 |
| | Male | 28,476 | 55.4 |
| | Unknown | 3 | 0.0 |
| **Race/skin color** | | | |
| | White | 14,970 | 29.2 |
| | Non-white | 17,438 | 33.9 |
| | Unknown | 18,975 | 36.9 |
| **Signs and Symptoms** | | | |
| | Fever | 30,466 | 59.3 |
| | Cough | 31,449 | 61.2 |
| | Odynophagia | 6,118 | 11.9 |
| | Dyspnoea | 30,930 | 60.2 |
| | Respiratory distress | 23,890 | 46.5 |
| | Oxygen saturation less than 95% | 26,286 | 51.2 |
| | Diarrhoea | 5,687 | 11.1 |
| | Vomiting | 3,335 | 6.5 |
| **Presence of risk factor/comorbidity** | | 32,616 | 63.5 |
| | Pregnancy | 267 | 0.5 |
| | Parturient | 134 | 0.3 |
| | Cardiovascular disease | 19,080 | 37.1 |
| | Diabetes mellitus | 12,578 | 24.5 |
| | Chronic kidney disease | 2,150 | 4.2 |
| | Chronic neurological disease | 1,872 | 3.6 |
| | Chronic lung disease | 1,640 | 3.2 |
| | Obesity | 2,127 | 4.1 |
| | Asthma | 957 | 1.9 |
| | Immunosuppression | 1,146 | 2.2 |
| | Hematological disease | 445 | 0.9 |
| | Chronic liver disease | 334 | 0.7 |
| **Use of antiviral** | | 6,203 | 12.1 |
| **Hospitalization** | | 47,673 | 92.8 |
| **Use of ICU** | | 19,500 | 38.0 |
| **Use of ventilatory support** | | | |
| | Without support | 10,302 | 20.0 |
| | Invasive | 8,215 | 16.0 |
| | Non-invasive | 20,185 | 39.3 |
| | Unknown | 12,681 | 24.7 |
| **Imaging Exam** | | | |
| | Normal | 661 | 1.3 |

*(Continued)*

**Table 1.** (Continued)

| Variable | | COVID-19 SARS (n = 51,383) | |
|---|---|---|---|
| | | n | % |
| | Atypical COVID-19 | 11,114 | 21.6 |
| | Typical COVID-19 | 14,814 | 28.8 |
| | Not performed/Unknown | 24,794 | 48.3 |
| **Criterion** | | | |
| | Laboratory | 39,863 | 77.6 |
| | Clinical-epidemiological | 398 | 0.7 |
| | Clinical/clinical-imaging | 7,640 | 14.8 |
| | Not closed | 3,544 | 6.9 |
| **Evolution** | | | |
| | Death | 20,785 | 40.5 |
| | No death | 30,598 | 59.5 |

SARS, severe acute respiratory syndrome; COVID-19, coronavirus disease 2019.

According to Yang *et al.* [5], Docherty *et al.* [6], and Huang et al. [7], COVID-19 was more frequent in men than in women, as in the present study. However, a study conducted in a federal hospital in Rio de Janeiro showed that 52.9% of the patients hospitalized for COVID-19 were women [8]. In addition, older patients and patients with previous diseases are more susceptible to SARS-CoV-2, which may explain the high frequency of comorbidities among patients with COVID-19 [5].

Bastos et al. [9] reported that the population aged over 60 years was the most severely affected by COVID-19 in Brazil. Other studies presented similar results, indicating that SARS-CoV-2 infection has substantially impacted the population over 60 years of age, as seen in the UK, with an average of 73 years [6] and in New York, with an average of 63 years [10]. In Spain, the average age was slightly lower at 57 years [11].

The most common clinical signs and symptoms reported in the literature are fever, cough, myalgia, dyspnoea, and fatigue [5,7,10,12]. Another clinical feature found in the studies was low partial oxygen saturation. For example, in a study conducted in Wuhan, patient oxygen saturation ranged from 84.9% to 95% [5]; another study conducted in New York reported that 20.4% of included patients had oxygen saturation below 90% [10]. The results mentioned above are similar to those found in the present study.

As in the present study, a previous study found the most relevant factors associated with death were the presence of comorbidities, including heart diseases and diabetes mellitus [13]. In another study, cardiovascular diseases (23.7%) and diabetes mellitus (10.3%) were the most common chronic diseases in patients admitted for COVID-19 [14].

Cardiovascular diseases and diabetes mellitus have been identified as significant risks for morbidity and mortality due to COVID-19 [15,16]. In a study by Borobia *et al.* (2020) [17], hypertension, chronic cardiovascular diseases, and diabetes mellitus had prevalence rates of 41%, 19%, and 17%, respectively. In addition, a Spanish study reported that hypertension was the main comorbidity among patients infected with SARS-CoV-2 [11].

In a study of patients hospitalized in a federal hospital in Rio de Janeiro, COVID-19 was associated with high mortality concerning discarded cases [8]. Similarly, a study conducted in the Northern Region of Brazil reported a significant difference between the mortality of confirmed and unconfirmed COVID-19, where COVID-19 showed higher mortality than other flu-like diseases [18].

**Table 2. Odds ratio and 95% confidence intervals of the covariates.**

| Variable | | Odds Ratio | 95%CI | P-value | C-Statistic |
|---|---|---|---|---|---|
| **Age group** | 18–49 years old (reference) | - | - | - | 0.676 |
| | 50–64 years old | 2.236 | 2.111–2.368 | 0.000 | |
| | 65–74 years old | 4.001 | 3.772–4.243 | 0.000 | |
| | 75 years or older | 6.394 | 6.039–6.77 | 0.000 | |
| **Sex** | Female (reference) | - | - | - | 0.500 |
| | Male | 0.995 | 0.96–1.031 | 0.780 | |
| **Race/skin color** | White (reference) | - | - | - | 0.502 |
| | Non-white | 0.986 | 0.944–1.031 | 0.540 | |
| **Signs and Symptoms** | Fever (reference: absence) | 0.752 | 0.716–0.79 | 0.000 | 0.525 |
| | Cough (reference: absence) | 0.773 | 0.735–0.813 | 0.000 | 0.521 |
| | Odynophagia (reference: absence) | 0.719 | 0.676–0.764 | 0.000 | 0.528 |
| | Dyspnoea (reference: absence) | 1.954 | 1.851–2.063 | 0.000 | 0.552 |
| | Respiratory distress (reference: absence) | 1.858 | 1.77–1.95 | 0.000 | 0.564 |
| | Oxygen saturation less than 95% (reference: absence) | 2.368 | 2.25–2.492 | 0.000 | 0.581 |
| | Diarrhoea (reference: absence) | 0.72 | 0.675–0.767 | 0.000 | 0.527 |
| | Vomiting (reference: absence) | 0.855 | 0.791–0.924 | 0.000 | 0.509 |
| **Presence of risk factor/comorbidity** | At least one risk factor/comorbidity (reference: absence) | 1.363 | 1.313–1.414 | 0.000 | 0.535 |
| | Pregnancy (reference: absence) | 0.278 | 0.201–0.385 | 0.000 | 0.503 |
| | Parturient (reference: absence) | 0.424 | 0.282–0.636 | 0.000 | 0.501 |
| | Cardiovascular disease (reference: absence) | 1.168 | 1.126–1.211 | 0.000 | 0.518 |
| | Diabetes mellitus (reference: absence) | 1.32 | 1.267–1.375 | 0.000 | 0.526 |
| | Chronic kidney disease (reference: absence) | 2.185 | 2.001–2.385 | 0.000 | 0.516 |
| | Chronic neurological disease (reference: absence) | 1.884 | 1.717–2.068 | 0.000 | 0.511 |
| | Chronic lung disease (reference: absence) | 1.864 | 1.688–2.058 | 0.000 | 0.510 |
| | Obesity (reference: absence) | 0.981 | 0.898–1.072 | 0.672 | 0.500 |
| | Asthma (reference: absence) | 0.664 | 0.579–0.762 | 0.000 | 0.503 |
| | Immunosuppression (reference: absence) | 1.334 | 1.187–1.5 | 0.000 | 0.503 |
| | Hematological disease (reference: absence) | 1.182 | 0.979–1.426 | 0.081 | 0.501 |
| | Chronic liver disease (reference: absence) | 1.568 | 1.264–1.944 | 0.000 | 0.501 |
| **Use of antiviral** | Yes (reference: no) | 1.507 | 1.423–1.596 | 0.000 | 0.536 |
| **Hospitalization** | Yes (reference: no) | 1.489 | 1.305–1.698 | 0.000 | 0.504 |
| **Use of ICU** | Yes (reference: no) | 2.101 | 2.016–2.189 | 0.000 | 0.592 |
| **Use of ventilatory support** | Without support (reference) | - | - | - | 0.709 |
| | Invasive | 12.938 | 12.061–13.879 | 0.000 | |
| | Non-invasive | 1.725 | 1.629–1.828 | 0.000 | |
| **Imaging Exam** | Normal (reference) | - | - | - | 0.573 |
| | Atypical COVID-19 | 1.76 | 1.48–2.094 | 0.000 | |
| | Typical COVID-19 | 0.971 | 0.816–1.154 | 0.737 | |
| **Criterion** | Laboratory (reference) | - | - | - | 0.548 |
| | Clinical-epidemiological | 1.629 | 1.337–1.985 | 0.000 | |
| | Clinical/clinical-imaging | 2.003 | 1.906–2.105 | 0.000 | |

Concerning the mortality profile of COVID-19 SARS, Cobre *et al.* [19] (2020) reported that male patients progressed to death more frequently than female patients. These authors also found a significant association between death from the disease and older age: patients aged 20–29 years died less frequently than those aged 70–79 and 80–89 years.

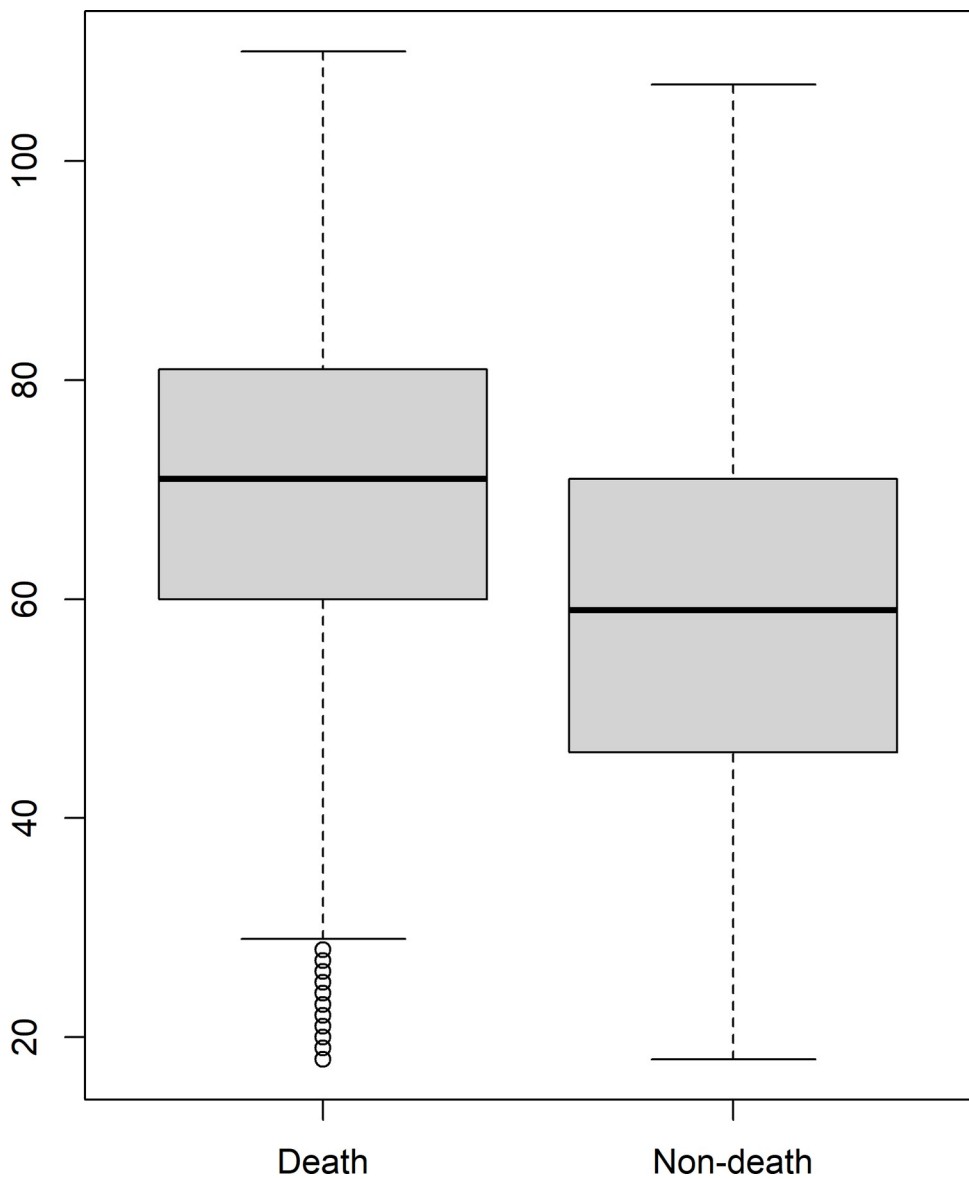

**Fig 1. Age distribution of SARS due to COVID-19 cases according to outcome (death/non-death), state of Rio de Janeiro, March–December 2020.**

A study on the clinical characteristics of individuals hospitalized for COVID-19 in Italy reported that patients had many comorbidities, the main ones being hypertension, diabetes mellitus, and ischemic heart disease. The prevalence of ischemic heart disease, atrial fibrillation, heart failure, stroke, hypertension, dementia, chronic obstructive pulmonary disease, and chronic kidney failure was significantly higher in patients aged 65 years or older than in patients younger than 65 years. In contrast, chronic obesity, liver disease, and HIV infection were significantly more frequent in younger patients than in older ones. The most prevalent comorbidities were cardiovascular disease, diabetes mellitus, cancer, dementia, and respiratory diseases among the patients who died. In line with previous studies, our data confirm that the presence of comorbidities is associated with a high risk of death in patients with COVID-19 [20].

**Table 3. Odds ratio and 95% confidence intervals of the covariates included in the final model.**

| Variable | | Odds Ratio | 95%CI | P-value | C-Statistic |
|---|---|---|---|---|---|
| **Age group (reference: 18–49 years old)** | 50–64 years old | 1.873 | 1.708–2.055 | 0.000 | 0.800 |
| | 65–74 years old | 3.148 | 2.863–3.462 | 0.000 | |
| | 75 years or older | 5.354 | 4.879–5.878 | 0.000 | |
| **Sex (reference: female)** | Male | 1.105 | 1.04–1.174 | 0.000 | |
| **Signs and Symptoms (reference: absence)** | Dyspnoea | 1.251 | 1.153–1.356 | 0.001 | |
| | Respiratory distress | 1.307 | 1.214–1.407 | 0.000 | |
| | Oxygen saturation less than 95% | 1.475 | 1.366–1.592 | 0.000 | |
| **Presence of risk factor/comorbidity (reference: absence)** | At least one risk factor/comorbidity | 1.320 | 1.232–1.415 | 0.000 | |
| | Chronic kidney disease | 1.941 | 1.688–2.233 | 0.000 | |
| | Chronic neurological disease | 1.362 | 1.175–1.579 | 0.000 | |
| | Immunosuppression | 1.508 | 1.257–1.809 | 0.000 | |
| **Use of ventilatory support (reference: without support)** | Invasive | 8.887 | 8.074–9.788 | 0.000 | |
| | Non-invasive | 1.247 | 1.151–1.351 | 0.000 | |

Arentz *et al.* [21] (2020) reported that chronic kidney diseases were the most frequent comorbidities (47.6%) in patients with COVID-19, followed by heart failure (42.9%). On the other hand, Docherty *et al.* (2020) [6] found an association between increased hospital mortality and chronic kidney disease, chronic lung, cardiac and neurological diseases, obesity, dementia, cancer, and liver diseases. These findings are similar to those presented in our study.

One main factor associated with death from COVID-19 in the present study was neurological disease. The same factor was reported in another study conducted in Espírito Santo, where

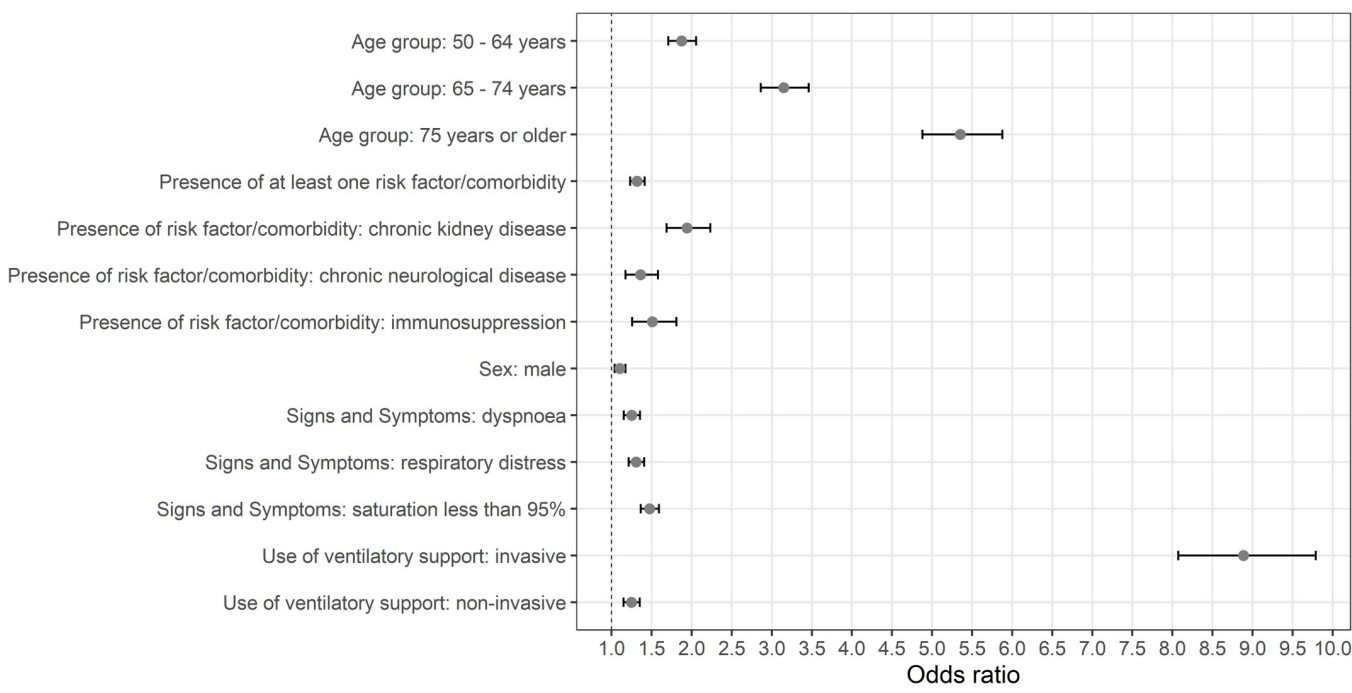

**Fig 2. Odds ratio and 95% confidence intervals of the covariates included in the final model.**

4.3% of patients hospitalized with COVID-19 had chronic neurological diseases and, of these, 88.9% died [13]. In addition, other studies have found that the presence of neoplasms [8,13,22,23], immunological diseases [13], and immunodeficiency [23] were predictors of death from COVID-19.

Studies conducted in the hospitals of Espírito Santo [13] and intensive care hospitals in England, Scotland, and Wales [6] found that COVID-19 was more lethal for obese patients than non-obese patients. Another study on the increased risk of hospitalization and death of patients with COVID-19 in Mexico showed that of the hospitalized patients, 23.52% were obese and, among the patients who died, 25% were obese [24]. The present study reported high mortality for patients on ventilatory support. Similar results were found by other authors [8,22,23,25,26].

Some limitations in the present study are related to the quality of information from epidemiological surveillance. The use of secondary data from information systems could be subject to underreporting and information bias.

## Conclusions

This study aimed to improve knowledge of the profile of adult patients with COVID-19 SARS in the state of Rio de Janeiro during the year 2020. In particular, the study was carried out to explain the severity of COVID-19 in Brazilian adults, as described in Oliveira et al. [26], and to understand the factors associated with death from the disease and better intervene in inpatient care, minimizing unfavorable outcomes and improving care for patients with COVID-19. COVID-19 SARS was associated with high mortality in our study. The main factors associated with death were male sex, old age, oxygen saturation <95%, respiratory distress, dyspnoea, chronic kidney and neurological diseases, immunosuppression, and the use of invasive and noninvasive ventilatory support.

## Supporting information

**S1 File.**
(CSV)

**S2 File.**
(DOCX)

## Author Contributions

**Conceptualization:** Tatiana de Araujo Eleuterio, Marcella Cini Oliveira, Roberto de Andrade Medronho.

**Data curation:** Tatiana de Araujo Eleuterio, Marcella Cini Oliveira, Carlos Eduardo Raymundo.

**Formal analysis:** Tatiana de Araujo Eleuterio, Marcella Cini Oliveira, Marlos Melo Martins, Roberto de Andrade Medronho.

**Investigation:** Tatiana de Araujo Eleuterio, Marcella Cini Oliveira, Mariana dos Santos Velasco, Rachel de Almeida Menezes, Regina Bontorim Gomes, Marlos Melo Martins, Roberto de Andrade Medronho.

**Methodology:** Tatiana de Araujo Eleuterio, Marcella Cini Oliveira.

**Project administration:** Tatiana de Araujo Eleuterio, Roberto de Andrade Medronho.

**Software:** Tatiana de Araujo Eleuterio, Marcella Cini Oliveira, Carlos Eduardo Raymundo.

**Supervision:** Tatiana de Araujo Eleuterio, Roberto de Andrade Medronho.

**Validation:** Tatiana de Araujo Eleuterio.

**Visualization:** Tatiana de Araujo Eleuterio, Marcella Cini Oliveira, Mariana dos Santos Velasco, Rachel de Almeida Menezes, Regina Bontorim Gomes, Roberto de Andrade Medronho.

**Writing – original draft:** Tatiana de Araujo Eleuterio, Marcella Cini Oliveira, Mariana dos Santos Velasco, Rachel de Almeida Menezes, Regina Bontorim Gomes, Roberto de Andrade Medronho.

**Writing – review & editing:** Tatiana de Araujo Eleuterio, Marcella Cini Oliveira, Mariana dos Santos Velasco, Rachel de Almeida Menezes, Regina Bontorim Gomes, Marlos Melo Martins, Roberto de Andrade Medronho.

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
