## [Decision Letter · Decision Letter 0]

4 Aug 2021

PONE-D-21-19718

SARS due to COVID-19: predictors of death and profile of patients in the state of Rio de Janeiro, 2020

PLOS ONE

Dear Dr. Eleuterio,

Thank you for submitting your manuscript to PLOS ONE. After careful consideration, we feel that it has merit but does not fully meet PLOS ONE’s publication criteria as it currently stands. Therefore, we invite you to submit a revised version of the manuscript that addresses the points raised during the review process.

We look forward to receiving your revised manuscript.

Kind regards,

Giordano Madeddu

Academic Editor

PLOS ONE

2. You indicated that ethical approval was not necessary for your study. We understand that the framework for ethical oversight requirements for studies of this type may differ depending on the setting and we would appreciate some further clarification regarding your research. Could you please provide further details on why your study is exempt from the need for approval and confirmation from your institutional review board or research ethics committee (e.g., in the form of a letter or email correspondence) that ethics review was not necessary for this study? Please include a copy of the correspondence as an "Other" file.

In addition, please ensure it is clear in your ethics statement that the Research Ethics Committee of the University Hospital Clementino Fraga Filho specifically granted an exemption, rather than an approval.

“This study was funded by the Carlos Chagas Filho Foundation for Research Support of the State of Rio de Janeiro - FAPERJ. Process E_18/2015TXB: AÇÃO EMERGENCIAL COVID-19 - Chamada A - Apoio a Rede de Pesquisa em Vírus Emergentes e Reemergentes. Rio de Janeiro, RJ, Brazil.

The National Council for Scientific and Technological Development (CNPq) and the Federal University of Rio de Janeiro (UFRJ) are responsible for granting the Scientific Initiation scholarship to co-author Marcella Cini Oliveira, through the Institutional Program for Scientific Initiation Scholarships (PIBIC).

The National Council for Scientific and Technological Development (CNPq) and Department of Training and Support for the Formation of Human Resources (DCARH) of the State University of Rio de Janeiro (UERJ) are responsible for granting the Scientific Initiation Scholarship by the Institutional Program of Scientific Initiation Scholarships (PIBIC) to the co-author Mariana dos Santos Velasco.”

We note that you have provided funding information that is currently declared in your Funding Statement. However, funding information should not appear in the Funding section or other areas of your manuscript. We will only publish funding information present in the Funding Statement section of the online submission form.

 “This study was funded by the Carlos Chagas Filho Foundation for Research

Support of the State of Rio de Janeiro (http://www.faperj.br/) FAPERJ. Process E_18/2015TXB: AÇÃO EMERGENCIAL COVID-19 - Chamada A - Apoio a Rede de Pesquisa em Vírus Emergentes e Reemergentes. Rio de Janeiro, RJ, Brazil.

The National Council for Scientific and Technological Development (CNPq) (https://www.gov.br/cnpq/pt-br) and the Federal University of Rio de Janeiro (UFRJ) are responsible for granting the Scientific Initiation scholarship to co-author MCO, through the Institutional Program for Scientific Initiation Scholarships (PIBIC).

The National Council for Scientific and Technological Development (CNPq) (https://www.gov.br/cnpq/pt-br) and the Department of Training and Support for the Formation of Human Resources (DCARH) of the State University of Rio de Janeiro (UERJ) are responsible for granting the Scientific Initiation Scholarship by the Institutional Program of Scientific Initiation Scholarships (PIBIC) to the co-author MSV.

The funders did not play any role in the study design, data collection and analysis, decision to publish, or preparation of the manuscript”

Reviewers' comments:

Reviewer's Responses to Questions

**Comments to the Author**

1. Is the manuscript technically sound, and do the data support the conclusions?

Reviewer #1: Yes

Reviewer #2: Partly

2. Has the statistical analysis been performed appropriately and rigorously? 

Reviewer #1: No

Reviewer #2: Yes

3. Have the authors made all data underlying the findings in their manuscript fully available?

Reviewer #1: Yes

Reviewer #2: Yes

4. Is the manuscript presented in an intelligible fashion and written in standard English?

Reviewer #1: Yes

Reviewer #2: Yes

5. Review Comments to the Author

Reviewer #1: Title: SARS due to COVID-19: predictors of death and profile of patients in the state of Rio de Janeiro, 2020.

Manuscript #: PONE-D-21-19718.

The contribution of this article is the identification of independent risk factors for death among COVID-19 cases in a population of the state of Rio de Janeiro in Brazil. The authors captured measures for this study and classified the variables into four domains: Demographic, clinical, epidemiological, and health care. Before going into my comments, I am going to describe how this analysis should have been carried out using the non-parametric modeling:

The primary aim of this study, per the authors, are

(i) to describe the clinical and epidemiological profile of patients with SARS in the state of Rio de Janeiro,

(ii) determine the predictors of death due to COVID-19-related SARS, and (iii) identify whether mortality patterns vary according to sociodemographic, clinical-epidemiological, and health care variables.

It is clearly defined the population for the study is for all COVID-19 cases. The logistic regression modeling should be focused on finding the independent factors associated with each domain and combine all the domains in the final model to see what are predictors of the outcome among the COVID-19 cases. More importantly, the study includes age less than 18 years, which will bias study estimates, for example, comparing 0-9 years versus greater than 80 years of age. Several research publications were out on those cohorts as the pandemic evolved and did more research on hospitalized adult patients with COVID-19. Including patients, less than 18 years in this study is not a correct approach. As reported by the authors in Table 3, only 2% of the cases are among them. I do not know what proportion of death among those two percent in the less than 18 years of age.

Comments:

1. Focus on COVID-19 cases. Including non-COVID-19 patients data divert the attention of the reader. Therefore, there is no need to include non-COVID-19 patients.

2. There should be a clear flow of data for an analyzable sample (Figure 1).

3. Exclude all data with less than 18 years of age.

4. The authors should follow the classification age per CDC standard; It should be age 18-49, 50-64, 65-74, and greater than or equal to 75 years.

5. Table 1 should be a distribution of all analyzable subjects among the measures in those four domains.

6. Table 2 should be odds ratios, 95% CI, and the associated probability value within each domain- report only the significant variables within each domain.

7. Table 3 should combine all the domain measures, use a step-wise logistic regression method to identify the independent factors associated with the outcome. Report the significant measures odds ratio, 95% CI, and its probability value.

8. Use the Table 3 odds ratio and its 95% CI to create a graphical representation as Figure 2.

Minor comments:

1. The current variables listed in table 1 be moved to the methods section, not as a table.

2. All the other Tables 2,3 and 4 are to be removed and modified as mentioned above in the comments.

3. Availability of data and the ethical aspects section should be moved to the methods section.

4. All the tables and figures should be moved to the end of the manuscript and provide clear marking in the text where the text and figures should go.

5. The authors should mention the c-statistic of the individual domain model and the overall step-wise all domain model in the statistical methods section.

6. The details of AIC, VIF, and ROC are unnecessary and should be deleted in the statistical methods section. Also, it is not essential to know those values in the results or discussion section.

A complete overhaul of analysis is needed to make it a tremendous COVID-19 research report from this retrospective data from Brazil.

Reviewer #2: Title: SARS due to COVID-19: predictors of death and profile of patients in the state of Rio de Janeiro, 2020 Manuscript number======PONE-D-21-19718

Review by Mastewal Arefaynie /Assistant professor in public health)

Wollo University

Dessie, Ethiopia

General comment

There are several topological and grammar usage errors that need extensive proof reading for revisions.

Specific comments

Abstract

1. In the introduction part you state simply the objective of the study. But it needs the justification of the research (the identified gap).

2. Material and method part it is enough to say method. So remove material. Try to include the software you used for analysis and the type logistic regression you were used.

3. Result: are you using all SARS case or COVID-19 patients only? Try to focus only on the latter case.

4. Line “32” comorbidity was risk factor for death. But you state comorbidity like kidney disease in line “32-34”. But preferred to use each commodity factor by removing there commorbidity effect.

5. Immunodepression change to immunosuppression

6. The conclusion part Line “36” by using odds ratio, you try to conclude to factors. But it is advisable to include more predictors with direction.

Introduction

Generally it is good. But need some justification.

7. The justification to do the research is not well described for scholars.

Methods

8. Change “Materials and Methods” to “methods”

9. “Line 82”, immediate notification of SARS cases was. Change cases to case or was to were.

10. “Line 86” during your description of the outcome variable, you say none death for hospitalization, and hospitalization in an intensive care unit patients. Why not declare their outcome? Unless miss-classification may be there. Your dependent variable should be cure and death.

11. You are using general linear logistic regression. But your outcome variable is dichotomous so binary logistic regression is appropriate. Also your result is expressed in OR.

Result

Unless you were doing comparative study among COVID-19 SARS and SARS, write the result only for you interest. Or compare them.

Discussion

Needs justification for factors

Wish you the luckiest!

Mastewal Arefaynie.

6. PLOS authors have the option to publish the peer review history of their article (what does this mean?). If published, this will include your full peer review and any attached files.

Reviewer #1: No

Reviewer #2: No

---

## [Author Response · Author response to Decision Letter 0]

21 Sep 2021

Accepted request. We use PLOS ONE's style requirements, including those for file naming.

2. You indicated that ethical approval was not necessary for your study. We understand that the framework for ethical oversight requirements for studies of this type may differ depending on the setting and we would appreciate some further clarification regarding your research. Could you please provide further details on why your study is exempt from the need for approval and confirmation from your institutional review board or research ethics committee (e.g., in the form of a letter or email correspondence) that ethics review was not necessary for this study? Please include a copy of the correspondence as an "Other" file.

In addition, please ensure it is clear in your ethics statement that the Research Ethics Committee of the University Hospital Clementino Fraga Filho specifically granted an exemption, rather than an approval.

Accepted request. We clarify in the ethical statement described in the last paragraph of the Methods section and the manuscript submission platform that the Research Ethics Committee of the University Hospital Clementino Fraga Filho decided for an exemption rather than an approval. 

We also included a letter from the Research Ethics Committee confirming that ethics review was not necessary for this study. It was included as an “Other” file.

We present below the translation of the decision of the Research Ethics Committee of the University Hospital Clementino Fraga Filho:

“DECISION DATA

Decision Number: 3,981,744

Project presentation:

Protocol 089-20 received on 04/18/2020.

Research Objective:

Not applicable.

Risk and Benefit Assessment:

Not applicable.

Research Comments and Considerations:

Not applicable.

Considerations for Mandatory Submission Terms:

Not applicable.

Recommendations:

Not applicable.

Conclusions or Pending Issues and List of Inadequacies:

Considering the provisions of Resolution CNS 510/2016, in its Article 1, paragraph one, item V, "The CEP/CONEP system shall not register or evaluate: V - research with databases, whose information is aggregated, without the possibility of identification individual”, it is understood that the research project mentioned above does not require ethical assessment by the CEP/Conep system.”

3. We note that you have provided funding information that is currently declared in your Funding Statement. However, funding information should not appear in the Funding section or other areas of your manuscript. We will only publish funding information present in the Funding Statement section of the online submission form.

Please remove any funding-related text from the manuscript and let us know how you would like to update your Funding Statement. 

Accepted request. We have removed funding information from the manuscript.

We want to modify the information in our funding statement; as we have informed you above, we have not received funding from any funding agency to carry out this research. Its development counted on the infrastructure of the Federal University of Rio de Janeiro and the scholarship of researcher Roberto de Andrade Medronho and scientific initiation scholarship holders Marcella de Oliveira Cini and Mariana dos Santos Velasco. We attach documents that prove the situation described above. Therefore, we request that our financing statement be updated as described below:

The Carlos Chagas Filho Foundation for Research Support of the State of Rio de Janeiro (http://www.faperj.br/) FAPERJ was responsible for granting a researcher scholarship to co-author RAM, Process E_18/2015TXB: AÇÃO EMERGENCIAL COVID-19 - Chamada A - Apoio a Rede de Pesquisa em Vírus Emergentes e Reemergentes. Rio de Janeiro, RJ, Brazil.

The National Council for Scientific and Technological Development (CNPq) (https://www.gov.br/cnpq/pt-br) and the Federal University of Rio de Janeiro (UFRJ) are responsible for granting the Scientific Initiation Scholarship to co-author MCO, through the Institutional Program for Scientific Initiation Scholarships (PIBIC).

The National Council for Scientific and Technological Development (CNPq) (https://www.gov.br/cnpq/pt-br) and the Department of Training and Support for the Formation of Human Resources (DCARH) of the State University of Rio de Janeiro (UERJ) are responsible for granting the Scientific Initiation Scholarship to co-author MSV, through the Institutional Program of Scientific Initiation Scholarships (PIBIC).

Accepted request. The ethics statement is only in the last paragraph of the Methods section.

Response to reviewers

Reviewer #1: 

Comments:

1. Focus on COVID-19 cases. Including non-COVID-19 patients data divert the attention of the reader. Therefore, there is no need to include non-COVID-19 patients.

Accepted request. The former Table 2 was reformulated as Table 1, showing only the profile of cases of SARS due to COVID-19. Table 2, which presented the comparison between SARS due to COVID-19 and non-COVID-19 SARS, was excluded.

2. There should be a clear flow of data for an analyzable sample (Figure 1).

Accepted request. We redid Figure 1 with the sample of COVID-19 cases in individuals over 18 years of age.

3. Exclude all data with less than 18 years of age.

Accepted request. We excluded all cases of COVID-19 in individuals under 18 years of age. Thus, we changed the title to: "SARS due to COVID-19: predictors of death and profile of adult patients in the state of Rio de Janeiro, Brazil, 2020".

4. The authors should follow the classification age per CDC standard; It should be age 18-49, 50-64, 65-74, and greater than or equal to 75 years.

Accepted request. We adopt age groups according to the CDC standard: 18-49, 50-64, 65-74, and 75 and older.

5. Table 1 should be a distribution of all analyzable subjects among the measures in those four domains.

Accepted request. We excluded the former Table 1. The description of the variables listed was for the Methods section (lines 91-95). The former Table 2 became Table 1 and presented only the SARS profile due to COVID-19. The text referring to the new Table 1 was reformulated with the new values (lines 138 – 147).

6. Table 2 should be odds ratios, 95% CI, and the associated probability value within each domain- report only the significant variables within each domain.

Accepted request. Table 2 presents the results of univariate logistic regressions for each covariate studied, considering the dependent variable death. We have updated the text referring to Table 2 (lines 152 – 160).

7. Table 3 should combine all the domain measures, use a step-wise logistic regression method to identify the independent factors associated with the outcome. Report the significant measures odds ratio, 95% CI, and its probability value.

Accepted request. Table 3 presents the results of the multivariate logistic regression for all significant covariates, considering the dependent variable death.

8. Use the Table 3 odds ratio and its 95% CI to create a graphical representation as Figure 2.

Accepted request. We created a graphical representation as Figure 2.

Minor comments:

1. The current variables listed in table 1 be moved to the methods section, not as a table.

Accepted request. We deleted the former Table 1 and insert the information from this table into the third paragraph of the Methods section (lines 92-96).

2. All the other Tables 2,3 and 4 are to be removed and modified as mentioned above in the comments.

Accepted request. We removed and modified tables 2, 3 and 4 as mentioned above in the comments.

3. Availability of data and the ethical aspects section should be moved to the methods section.

Accepted request. Statements about data availability and ethical aspects are described in the last two paragraphs of the Methods section.

4. All the tables and figures should be moved to the end of the manuscript and provide clear marking in the text where the text and figures should go.

Accepted request. We moved all tables and figures to the end of the manuscript and provided a clear mark where the text and figures should go.

5. The authors should mention the c-statistic of the individual domain model and the overall step-wise all domain model in the statistical methods section.

Accepted request. We entered the C-statistic for the individual models and for the final model in the Methods (rows 116-119), Results (178-179) and Tables 2 and 3 sections. 

6. The details of AIC, VIF, and ROC are unnecessary and should be deleted in the statistical methods section. Also, it is not essential to know those values in the results or discussion section.

Accepted request. We have removed the details about AIC, VIF, and ROC methods. 

A complete overhaul of analysis is needed to make it a tremendous COVID-19 research report from this retrospective data from Brazil.

We thank the reviewer for his comments and contributions.

Reviewer #2: Review by Mastewal Arefaynie /Assistant professor in public health), Wollo University

General comment

There are several topological and grammar usage errors that need extensive proof reading for revisions.

Accepted request. The manuscript was submitted for professional English language review.

Specific comments

Abstract

1. In the introduction part you state simply the objective of the study. But it needs the justification of the research (the identified gap).

Accepted request. We have added the research justification in lines 21-23.

2. Material and method part it is enough to say method. So remove material. Try to include the software you used for analysis and the type logistic regression you were used.

Accepted request. We changed the section title to “Methods”. In addition, we have included information about the software used for data analysis and the type of logistic regression (binary) in lines 25-28 of the Abstract. Furthermore, in lines 101-102 of the Methods section, we described the software used for data analysis, and in lines 113-114, we described the type of logistic regression (binary) used.

3. Result: are you using all SARS case or COVID-19 patients only? Try to focus only on the latter case.

Accepted request. We only analyzed SARS cases due to COVID-19.

4. Line “32” comorbidity was risk factor for death. But you state comorbidity like kidney disease in line “32-34”. But preferred to use each commodity factor by removing there commorbidity effect.

Accepted request. The variable “at least one risk factor/comorbidity” represents the fact that each individual has any and at least one comorbidity (considering all those listed individually in the model). Therefore, the variables “presenting at least one risk factor/comorbidity” were considered, as well as each comorbidity separately (pregnancy, parturient, cardiovascular disease, diabetes mellitus, chronic kidney disease, chronic neurological disease, chronic lung disease, asthma, asthma, immunosuppression, haematological disease, and chronic liver disease). We clarify this information in line 34 of the Abstract (“to present at least one risk factor/comorbidity”).

5. Immunodepression change to immunosuppression

Accepted request. We modified the term throughout the manuscript and tables.

6. The conclusion part Line “36” by using odds ratio, you try to conclude to factors. But it is advisable to include more predictors with direction.

Accepted request. In the “Conclusions” section of the Abstract, we included all predictors of death that had statistical significance (lines 39-41).

Introduction

Generally it is good. But need some justification.

7. The justification to do the research is not well described for scholars.

Accepted request. We insert a paragraph describing the justification for the research in lines 70-75.

Methods

8. Change “Materials and Methods” to “methods”

Accepted request. We changed the section title to “Methods”.

9. “Line 82”, immediate notification of SARS cases was. Change cases to case or was to were.

Accepted request. We corrected the sentence for “Immediate notification of SARS cases were carried out…” in line 91.

10. “Line 86” during your description of the outcome variable, you say none death for hospitalization, and hospitalization in an intensive care unit patients. Why not declare their outcome? Unless miss-classification may be there. Your dependent variable should be cure and death.

Accepted request. The dependent variable was defined as “death due to COVID-19 SARS or cure (non-death)”, in lines 96-97.

11. You are using general linear logistic regression. But your outcome variable is dichotomous so binary logistic regression is appropriate. Also your result is expressed in OR.

Accepted request. The information was corrected in lines 113-114, as binary logistic regression was used, not linear. We appreciate the error signaling by the reviewer.

Result

Unless you were doing comparative study among COVID-19 SARS and SARS, write the result only for you interest. Or compare them.

Accepted request. Initially, we were doing a comparative study between cases of SARS due to COVID-19 and SARS non-COVID. However, as we were asked by reviewer #1 to remove comparisons with SARS non-COVID-19, we complied with that suggestion. Therefore, the former Table 2 was reformulated as Table 1, showing only the SARS profile by COVID-19. Table 2, which presented the comparison between SARS due to COVID-19 and SARS non-COVID-19, was excluded.

Discussion

Needs justification for factors

Accepted request. We insert a paragraph on the justification of factors in lines 267-271.

---

## [Decision Letter · Decision Letter 1]

26 Oct 2022

SARS due to COVID-19: predictors of death and profile of adult patients in the state of Rio de Janeiro, 2020

PONE-D-21-19718R1

Dear Dr. Rodrigues de Araujo Eleuterio,

We’re pleased to inform you that your manuscript has been judged scientifically suitable for publication and will be formally accepted for publication once it meets all outstanding technical requirements.

Kind regards,

Paavani Atluri

Academic Editor

PLOS ONE

Additional Editor Comments (optional):

Reviewers' comments:

Reviewer's Responses to Questions

**Comments to the Author**

1. If the authors have adequately addressed your comments raised in a previous round of review and you feel that this manuscript is now acceptable for publication, you may indicate that here to bypass the “Comments to the Author” section, enter your conflict of interest statement in the “Confidential to Editor” section, and submit your "Accept" recommendation.

Reviewer #3: All comments have been addressed

2. Is the manuscript technically sound, and do the data support the conclusions?

Reviewer #3: Yes

3. Has the statistical analysis been performed appropriately and rigorously? 

Reviewer #3: Yes

4. Have the authors made all data underlying the findings in their manuscript fully available?

Reviewer #3: Yes

5. Is the manuscript presented in an intelligible fashion and written in standard English?

Reviewer #3: Yes

6. Review Comments to the Author

Reviewer #3: The present work aimed to analyze and describe predictors of death from severe acute respiratory syndrome (SARS) due to COVID-19 in Rio de Janeiro, Brazil, trought the System for Epidemiological Surveillance of Influenza and the Mortality Information. In general, the authors concluded that male sex, old age, oxygen saturation <95%, respiratory distress, dyspnoea, chronic kidney and neurological diseases, immunosuppression, and use of invasive or noninvasive ventilatory support were related to death.

As detailed in the file attached, the authors respond all the request made from previous reviewers.

That being said, and, due to the importance of manuscript regarding the issue raised, I reccomend the manuscrip for publication in PLOS ONE.

Best,

7. PLOS authors have the option to publish the peer review history of their article (what does this mean?). If published, this will include your full peer review and any attached files.

Reviewer #3: **Yes: **Raiane Cardoso Chamon
